# Exploring the Link Between Body Appreciation and Health-Related Lifestyle in Adolescents: A Cross-Sectional Study

**DOI:** 10.3390/bs15101400

**Published:** 2025-10-15

**Authors:** Migle Baceviciene, Laima Trinkuniene, Rasa Jankauskiene

**Affiliations:** 1Department of Physical and Social Education, Lithuanian Sports University, 44221 Kaunas, Lithuania; laima.trinkuniene@lsu.lt; 2Institute of Sport Science and Innovations, Lithuanian Sports University, 44221 Kaunas, Lithuania; rasa.jankauskiene@lsu.lt

**Keywords:** body image, health-related lifestyle, prevention, eating disorders, nutrition

## Abstract

Adolescence is a critical developmental stage at which body image and lifestyle behaviours intersect. Research shows that having a positive body image during this period is linked to better mental health and certain aspects of a healthy lifestyle. However, more empirical evidence is needed, especially concerning boys. This cross-sectional study explored the association between body appreciation (BA) and lifestyle factors in a large sample of Lithuanian adolescents. These associations were examined while controlling for body mass index (BMI), and the role of sex in these relationships was evaluated. A cross-sectional study involved 1412 adolescents (59.6% girls) aged 16–17 years (mean age of 16.97 ± 0.46 years). Participants completed questionnaires assessing BA, self-esteem, life satisfaction and lifestyle factors such as physical activity and perceived fitness, sleep duration, screen time and disordered eating (DE) attitudes and behaviours. Data were analysed using descriptive statistics, analysis of covariance (ANCOVA), and binary logistic regression analysis. Higher BA was associated with greater life satisfaction, self-esteem, perceived physical fitness and healthier eating behaviours independent of BMI in boys and girls. Adolescents with high BA had significantly higher odds of participating in sports, having a healthy BMI, accurate body weight estimation, good self-rated health and non-smoking behaviours compared to those with low BA. BA was also associated with healthier sleeping hours, lower screen time and lower unhealthy and DE behaviour in girls. Boys with high BA were more likely to abstain from alcohol. Interaction effects indicated that the effect of BA on self-esteem, BMI and DE behaviours was stronger in girls than in boys. BA is strongly linked to positive lifestyle outcomes and self-esteem in adolescents, particularly in girls. The findings of this study indicate that initiatives designed to promote healthy lifestyles among adolescent boys and girls may be enhanced by the incorporation of education on positive body image. Interventions should be tailored to gender-specific needs, emphasizing prevention of dysfunctional eating for girls, and reducing substance use for boys. Incorporating body-positive education into schools and health programs can help create supportive environments that enhance both psychological well-being and physical health.

## 1. Introduction

The health-related lifestyles of children and adolescents represent a critical public health concern. Adolescence is a period marked by significant physical, psychological, and social changes that influence growth and development ([72]) and these changes can make it challenging to develop or maintain a positive body image ([85]). Globally, multiple unhealthy lifestyle behaviours—including insufficient sleep and physical activity, sedentary behaviour, unhealthy eating, smoking, and alcohol consumption—are highly prevalent among adolescents, often accompanied by body image concerns ([23]; [86]). Understanding the environmental and individual factors that help prevent such behaviours is therefore a key public health and health promotion priority. The present study focuses on individual factors, specifically examining the relationship between positive body image and engagement in health-promoting behaviours among adolescent girls and boys.

Body image reflects how people perceive, what they think and feel and how they care about their bodies ([16]). The field of body image research has historically centred on the examination of adverse aspects of negative body image, with a particular emphasis on the phenomenon of body dissatisfaction ([31]). According to the sociocultural theory of negative body image ([76]), sociocultural pressures contribute to body dissatisfaction and self-objectification. These, in turn, lead to outcomes such as disordered eating, low self-esteem and distorted perceptions of body weight—overestimation in girls and underestimation in boys. These effects are largely driven by the internalisation of culturally promoted body ideals and comparisons with an idealised standard of beauty, with girls more likely to internalise ideals of thinness and boys more likely to internalise ideals of muscularity ([73]; [76]). A recent systematic review concluded that the prevalence of weight dissatisfaction in adolescents ranges from 18.0 to 56.6% in both sexes (10.8–82.5% among boys and 19.2–83.8% among girls) ([58]). Negative body image in adolescents has been constantly associated with poorer mental and physical health outcomes ([2]; [4]; [13]; [15]; [21]; [33]; [45]; [83]; [63]; [83]) and more negative lifestyle-related behaviours such as smoking, cannabis use, high-risk drinking, drug use and self-harm ([13]; [17]), screen-based activities ([60]), drive for muscularity ([14]) and disordered eating ([13]).

In last decades, there has been a marked increase in the study of positive body image ([1]; [53]). Positive body image has a novel and much broader conceptualisation as a multifaceted phenomenon and is a distinct construct from negative body image ([6]). The most extensively researched facet of positive body image is body appreciation (BA). BA is defined as acceptance, positive regard, love and respect for the body, and rejection of media-promoted appearance ideals as the sole form of beauty ([81]). The scientific community has recently been paying increased attention to the concept of positive body image (operating as BA) in lifestyle-related scientific literature, given its associations with positive behavioural, psychological and general well-being outcomes ([54], [53], [55]; [82]). However, little is known about how positive body image relates to health-related lifestyle factors in adolescents, particularly boys. Further exploration of these associations could yield new empirical data and valuable insights to inform evidence-based health promotion programmes targeting young people.

The Acceptance Model of Intuitive Eating suggests that accepting one’s body fosters BA and an internal body orientation. This, in turn, promotes intuitive eating—a self-care approach that emphasises responding to internal hunger and fullness cues rather than external dietary rules or restrictions ([7]). In line with the theory, research conducted with adolescent girls indicates that positive body image is associated with healthier dietary habits, a greater sense of autonomy in dietary management ([64]) and lower disordered eating ([57]; [66]). Other studies show that positive body image is related to self-perceived physical health and life satisfaction ([64]; [67]), engagement in sports and physical activity ([9]; [43]; [64]), and preventive sexual and cancer health behaviours ([61]). BA in girls has been found to be negatively associated with health-compromising behaviours including risky sexual activity and alcohol and tobacco use ([61]). A prospective study conducted on female adolescents demonstrated that a higher level of BA was associated with a reduced likelihood of dieting, alcohol consumption, and cigarette use, and an increased propensity for physical activity one year later. Specifically, girls who exhibited low BA were more susceptible to developing alcohol and cigarette use patterns between measurement points, indicating that fostering heightened BA could serve as a pivotal target for interventions aimed at curtailing or postponing the onset of alcohol and cigarette consumption among female adolescents ([3]).

Despite limited research on positive body image in boys, existing studies suggest that BA is linked to higher self-esteem ([40]; [50]), greater sports participation and physical activity ([41]; [43]; [69]), lower disordered eating ([9]; [66]) and reduced muscle dysmorphia ([66]) suggesting that positive body image might play a vital role in supporting both healthy lifestyles and overall well-being in boys. Thus, it is important to provide more empirical data on this issue.

Growing evidence links positive body image to healthier psychological and behavioral outcomes; however, most studies have focused on female samples or limited health behaviors, such as physical activity and eating. Much less is known about how positive body image relates to a broader range of health-related behaviors—such as smoking, alcohol use, sleep, and self-rated health—and whether these associations differ between adolescent girls and boys. Understanding these associations is important, as adolescence represents a critical period that shapes long-term habits, self-identity, academic engagement, and mental health ([12]; [28]; [49]; [89]). Generating new evidence on these associations is essential to advance understanding of the role of positive body image in adolescent health and to inform more inclusive and psychologically safe health promotion strategies.

At the same time, despite increasing evidence and initiatives promoting healthy lifestyles and well-being among adolescents of all body shapes and sizes, and efforts to foster inclusive educational environments, health promotion strategies still largely focus on body weight as the main target for changing health-related behaviour ([37]; [62]; [80]; [87]). This weight-centred approach can inadvertently reinforce weight stigma and associated concerns, which are linked to avoiding exercise, dieting and disordered eating behaviours ([11]; [33]; [52]; [56]; [59]). By contrast, a positive body image has been linked to healthier body mass ranges ([25]; [90]), greater intuitive eating among girls ([47]) and higher levels of physical activity in girls and boys ([41]). Therefore, it is crucial to gather more empirical evidence on potentially new, psychologically safer approaches to encouraging healthy lifestyles, especially among vulnerable groups such as adolescents. Generating more empirical evidence on the relationship between positive body image and health-related behaviours and well-being in both boys and girls, while accounting for body mass index (BMI), could inform more effective health education and support the development of gender-sensitive interventions.

Therefore, the present study aimed to comprehensively examine how BA—a core component of positive body image—is related to a range of health-related lifestyle factors and psychosocial well-being indicators in adolescent girls and boys. Specifically, we investigated whether higher BA is associated with more positive lifestyle behaviours (e.g., greater physical activity, healthier eating attitudes, longer sleep duration, less screen and gaming time, and lower smoking and alcohol use) and higher psychosocial well-being (self-esteem, life satisfaction, and self-rated health), controlling for body mass index (BMI). We further examined whether these associations differ between girls and boys. Based on previous findings, we hypothesized that adolescents with higher BA would report healthier behaviours, higher well-being, lower BMI, and more accurate body weight estimation.

## 2. Materials and Methods

### 2.1. Study Design and Procedure

The cross-sectional study was conducted in 2019–2020 in Lithuanian schools representing the major municipalities of the country. The study recruitment period was from 30 September 2019 to 31 January 2020. School principals were initially contacted to obtain permission to carry out the survey. The contacts were established based on prior collaborations or projects. Upon receiving approval, information about the study, including its aims and survey content, was communicated to the parents or caregivers of the students, requesting their consent for the participation of their children. If consent was not provided, the students were excluded from the survey.

The study was conducted during school breaks or after school hours under the supervision of physical education teachers. Before completing the survey, students were informed about the study’s aims, the content of the questions, and the survey duration (approximately 30 min). They were then asked to provide digital consent by selecting either “I agree to participate” or “I decline to participate”. Students who chose not to participate were acknowledged, and the survey was discontinued for them. The survey was anonymous and non-diagnostic, and participants were informed in advance that some questions might address sensitive issues such as body image or eating behaviours. Also, they were free to discontinue participation at any point without their responses being saved. Because the study was conducted with a general school-based adolescent population, no clinical follow-up or signposting to external support services was provided. Also, we did not apply exclusion criteria related to eating disorder diagnoses or symptomatology, as the study aimed to capture a broad and ecologically valid picture of adolescents’ positive body image and lifestyle factors.

The survey was administered via the SurveyMonkey platform, with all questions set as mandatory, resulting in a dataset with no missing values. The study was approved by the Committee for Social Sciences Research Ethics of the Lithuanian Sports University (protocol No. SMTEK-32).

### 2.2. Study Participants

The final sample comprised 1412 students with a mean age of 16.97 years (SD = 0.46). Of these, 570 (40.4%) were boys and 842 (59.6%) were girls. The inclusion criteria for the study required that the selected schools follow a general education curriculum and use Lithuanian as the language of teaching.

### 2.3. Study Measures

#### 2.3.1. Body Appreciation

The second version of the Body Appreciation Scale (BAS-2) ([81]) is a widely used psychological instrument designed to measure respondents’ acceptance, favourable opinions and respect of their own body. The BAS-2 consists of 10 items that evaluate how individuals respect their body and its functions; feel good about their body despite its imperfections; appreciate their body regardless of its shape, size or appearance; resist media and social pressures to conform to unrealistic beauty ideals; and engage in behaviours that reflect body care and appreciation. Each item is rated on a 5-point Likert scale, ranging from 1 (never) to 5 (always). Calculated higher mean scores reflect greater BA. The BAS-2 has shown high reliability and validity across diverse populations, including both men and women with a unidimensional factor structure and good internal consistency (Cronbach’s alpha is typically above 0.90). In the Lithuanian adolescent sample, the estimated psychometric properties of the BAS-2 were also good ([9]). For this study, Cronbach’s α was 0.97. As population norms for Lithuanian adolescents are not available for the BAS-2, for further analysis, the BAS-2 score was categorised separately for boys and girls to reflect low (−1 standard deviation [SD] from the sample), moderate (mean ± 1 SD) and high BA (+1 SD). Such score categorisation has also been presented in previous studies ([34]; [39]; [88]).

#### 2.3.2. Self-Esteem

Self-esteem was assessed using the Rosenberg Self-Esteem Scale (RSES), which evaluates an individual’s general self-esteem ([71]). The scale consists of 10 statements, with half phrased positively and the other half negatively. Responses are rated on a 4-point Likert scale, ranging from 1 (strongly disagree) to 4 (strongly agree). The negatively worded items are reverse-scored, and the total self-esteem score is calculated, ranging from 10 to 40, with higher scores reflecting greater self-esteem. The RSES demonstrated good internal consistency in this study, with a Cronbach’s alpha of 0.86. The RSES has been used in previous studies with adolescent samples ([9]; [42]).

#### 2.3.3. Life Satisfaction

Life satisfaction was measured by Cantril Ladder, as used in the international HBSC study ([36]). This instrument is a self-anchoring scale that allows respondents to rate their overall life satisfaction. It presents the visual of a ladder, where the bottom rung (0) represents the worst possible life, and the top rung (10) represents the best possible life.

#### 2.3.4. BMI and Body Weight Estimation

BMI was calculated as self-reported weight, kg/(height, m)^2^. Using the cut-offs suggested by the International Obesity Task Force (IOTF) for children and adolescents, study participants were categorised into underweight, healthy body weight, overweight and obese ([19]). The accuracy of body weight estimation was assessed by comparing participants’ self-reported current body weight and calculated BMI with their expressed opinion about their weight. Adolescents who were underweight or of healthy weight but perceived their weight as too high were classified in the overestimation group. Adolescents were categorised as having an accurate perception of their body weight if their self-assessment of being underweight, normal weight or overweight aligned with their actual weight. Finally, overweight or obese adolescents who perceived their weight as normal or too low were placed in the underestimation category. Such classification has been used in previous studies in adolescent samples ([35]; [68]).

#### 2.3.5. Sports and Physical Activity

Participation in sports was assessed by a single question asking the respondents to indicate if they are engaged in any sports or exercise activities after school with the response options “yes” or “no”. Physical activity was measured using the Godin and Shepard Leisure-Time Exercise Questionnaire (LTEQ) ([29]; [30]). This questionnaire includes three items, each providing examples of activities at different intensities: strenuous (e.g., running, jogging, football, basketball), moderate (e.g., brisk walking, casual cycling, badminton, light swimming) and light (e.g., yoga, archery, fishing, golf, leisurely walking). Participants were asked to report the approximate number of sessions they engaged in for each intensity level during the previous week, with each session lasting at least 15 min. If no sessions were performed at a given intensity, participants were asked to record zero. The number of sessions for strenuous, moderate and light activities was then multiplied by nine, five and three, respectively, and summed to produce a total score reflecting overall leisure-time physical activity level.

#### 2.3.6. Perceived Physical Fitness (PPF)

Perceived physical fitness (PPF) was assessed by a single question “How would you consider your physical fitness when comparing yourself to your peers?” with the response options 0—very weak; 1—a little weak; 2—average, like most peers; 3—a little strong; and 4—very strong. This question was designed to measure the subjective perception of one’s physical fitness relative to others, which can offer insight into self-assessed health and physical competence. The question has been used in previous studies ([40]).

#### 2.3.7. Self-Rated Health

Self-rated health was evaluated using the question “Would you say your health is …?” with possible answers 1—excellent; 2—good; 3—fair; and 4—poor. This question has previously been used in international lifestyle-related behaviours studies in adolescents ([36]).

#### 2.3.8. Sleep Duration

Sleep duration was assessed by a single item asking to indicate the average number of hours spent sleeping each day: “How many hours do you usually sleep per night?” Respondents could indicate the number of hours spent sleeping. This question has been used in previous studies assessing sleep in various samples ([70]).

#### 2.3.9. Screen-Based Activities

To assess time spent on computer gaming a single question was used: “On a typical day, how many hours do you spend playing computer or video games?” Participants were asked to estimate their daily gaming time, including games played on consoles, computers or mobile devices, with response options provided in hour increments (e.g., less than 1 h, 1 h, 2 h, 3 h, etc.). Another question was provided to evaluate screen time for browsing for non-educational purposes: “On a typical day, how many hours do you spend browsing the internet for non-educational purposes (e.g., social media, watching videos)?” Participants were instructed to report their daily time spent on non-educational internet activities, using the same hour increment scale. Similar questions have been used in previous studies ([78]).

#### 2.3.10. Eating Attitudes and Behaviours

The Eating Disorder Examination Questionnaire 6 (EDE-Q 6) is a widely used self-report measure designed to assess the core behavioural and attitudinal features of DE ([26]). The EDE-Q 6 consists of 28 items, which were originally grouped into four subscales: restraint evaluates attempts to restrict food intake; eating concern measures preoccupation with eating, guilt about eating and concerns about losing control over eating; shape concern assesses dissatisfaction with body shape, fear of gaining weight and the influence of shape on self-worth; and weight concern evaluates dissatisfaction with body weight, preoccupation with weight and the influence of weight on self-worth. EDE-Q 6 items are rated on a 7-point Likert scale, ranging from 0 (no days/not at all) to 6 (every day/markedly), with higher scores indicating more severe DE symptoms. The questionnaire also includes six items assessing key ED behaviours, such as episodes of binge eating, self-induced vomiting, excessive exercise to affect body weight and the use of laxatives or diuretics. These items are reported as frequencies of episodes in the past 28 days. The EDE-Q 6 also provides a global score, which is the average of the four subscale scores. The global score used in this study gives an overall measure of DE. The psychometric properties of the Lithuanian translation of the EDE-Q 6 have been previously confirmed ([8]), and in this study, Cronbach’s α was 0.95.

#### 2.3.11. Unhealthy Eating Habits

Unhealthy eating habits were evaluated using a five-item tool developed for the Health Behaviour Survey among the Lithuanian Adult Population ([44]). This tool assesses the frequency of specific eating behaviours, such as overeating, eating in a rush, eating while watching TV, consuming unhealthy snacks like chips or sweets and eating late at night (less than 2 h before sleep). Responses are rated on a 5-point Likert scale, ranging from 0 (never) to 4 (always). The calculated average of the response options indicates more frequent unhealthy eating behaviour. This scale has been used in previous studies with young Lithuanians ([10]).

#### 2.3.12. Smoking and Alcohol Consumption

Current smoking behaviour was assessed using a single question from the HBSC study: “How often do you smoke cigarettes at present?” The response options were “I do not smoke”, “less than once a week”, “at least once a week, but not every day” and “every day”. For further analyses, individuals who reported current smoking at least once a week were classified as smokers. These questions were used previously in HBSC study ([17]).

The frequency of risky alcohol consumption leading to drunkenness was assessed with a single question: “Have you ever had so much alcohol that you were really drunk?” Response options included: “never”, “once”, “2–3 times”, “4–10 times” and “more than 10 times”. Adolescents who reported being drunk 2–3 times or more were classified as risky alcohol consumers. These questions were used in the HBSC study ([36]).

### 2.4. Statistical Analysis

The sample size of 1412 adolescents provided adequate statistical power (≥0.80) to detect small to medium effect sizes (partial η^2^ = 0.01–0.06) in ANCOVA and odds ratios of 1.5–2.0 in logistic regression models at α < 0.05 ([18]). Initial descriptive statistics, normality testing of continuous variables and calculation of Cronbach’s α were performed. Sample characteristics were presented as the number of study participants in the categories of the variables and the percentage. Continuous variables were presented as means with standard deviations. Next, analysis of covariance (ANCOVA) was conducted to compare the means of the continuous study variables (self-esteem, physical activity score, DE score calculated from the EDE-Q 6, etc.) in three groups of BA as a fixed factor, with BMI was entered as a covariate; the analysis was run separately for boys and girls. We applied post hoc analyses and controlled for multiple comparisons to reduce the risk of Type I error. Partial eta-squared was calculated to assess the effect size for significant differences and was classified as small (above 0.01 and below 0.06), medium (above 0.06 and below 0.14) and large (≥0.14) ([18]). The interaction effects between sex and categorised BA on continuous study variables were then tested. Finally, binary logistic regression was run to predict healthy lifestyle factors (healthy BMI, good/excellent self-rated health, non-smoking, abstaining from alcohol, etc.) by BA with the lowest BA set as a reference group. In all models, BMI was entered as a covariate. Also, the interaction effects between sex and BA on healthy lifestyle habits were tested. Statistical analysis was performed using the free software JASP v.0.18.3 obtained from the official website (JASP Team: Amsterdam, The Netherlands). A *p*-value below 0.05 was considered statistically significant.

## 3. Results

Table 1 provides a comprehensive description of the sample’s characteristics, highlighting generally positive BA, self-esteem and life satisfaction. Most participants in the sample demonstrated a healthy BMI, engaged in sports, scored 2.10 ± 1.01 for their perceived physical fitness in a range of 0 to 4 and reported good or excellent self-rated health. Study participants spent about an hour on computer gaming and about 4 h on non-educational internet browsing a day. There were quite low scores for unhealthy eating habits (1.76 ± 0.63 in a range from 0 to 4) and DE (1.38 ± 1.24 in a range from 0 to 6) in the sample. Smoking and alcohol consumption until feeling dizzy appeared at different levels. The prevalence of smoking at least once a week was 21.5%, but the prevalence of taking alcohol until feeling dizzy more than once was 50.2%.

Table 2 and Table 3 present the ANCOVA models for adolescent boys and girls separately comparing different variables across low-, moderate- and high-BA groups with the BMI entered as a covariate (except when the dependent variable was BMI). The results showed that boys with higher BA were more satisfied with their lives and had higher levels of self-esteem and perceived physical fitness and lower scores of DE behaviours independent of their BMI. The effect sizes (partial ŋ2) were small to moderate.

In girls, higher BA was associated with better outcomes across most analysed factors, including higher life satisfaction, self-esteem, perceived physical fitness and lower levels of DE behaviours (Table 3). Lower BA was linked to higher BMI, poorer self-esteem, more expressed DE attitudes and behaviours, as well as less life satisfaction. The effect sizes (partial ŋ^2^) indicated that self-esteem, life satisfaction and DE had the strongest associations with BA levels. At the same time, other factors like physical fitness perception, sleep duration, screen-time behaviour, eating attitudes and behaviours showed smaller, yet still significant associations, independent of BMI. Only leisure-time exercise scores did not differ across BA groups in adolescent girls.

Next, the interaction effects between sex and BA groups on the dependent variables presented before were also tested. The analysis revealed significant interaction effects on self-esteem (F = 7.9, ŋ^2^ = 0.01, *p* < 0.001), BMI (F = 4.3, ŋ^2^ = 0.006, *p* = 0.013) and DE behaviours (F = 31.8, ŋ^2^ = 0.04, *p* < 0.001); the effects of BA on these variables were stronger in girls (on self-esteem F = 155.4, ŋ^2^ = 0.27, *p* < 0.001, on BMI F = 27.4, ŋ^2^ = 0.06, *p* < 0.001, and on DE F = 103.4, ŋ^2^ = 0.20, *p* < 0.001, respectively) than in boys (on self-esteem F = 41.9, ŋ^2^ = 0.13, *p* < 0.001, on BMI F = 2.4, ŋ^2^ = 0.008, *p* = 0.092, and on DE F = 10.1, ŋ^2^ = 0.04, *p* < 0.001, respectively).

Table 4 reveals that BA has a significant association with various healthy lifestyle factors when controlling for BMI. In both boys and girls, higher BA was associated with higher odds of having a healthy BMI, accurate body weight estimation versus overestimation, better self-rated health and higher participation in sports. Importantly, when predicting adequate body weight estimation versus underestimation, no significant differences were found in boys and girls, so the findings are not included in the table. Boys with higher BA were also more likely to abstain from smoking and alcohol, while girls in the group with the highest BA only had higher odds of non-smoking compared to the lowest BA. For most lifestyle factors, the effect of high BA was stronger in girls than in boys, especially regarding body weight estimation, healthy BMI and self-rated health. The associations were statistically significant in many cases, especially at higher levels of BA. Moreover, when predicting a healthy BMI, a significant sex and BA interaction effect was observed (*p* = 0.018), which indicates that higher BA for girls had a significantly stronger effect on meeting the recommendations of healthy BMI (OR = 2.40, 95% CI = 1.50–3.87, *p* < 0.001 in moderate BA and OR = 8.78, 95% CI = 3.35–23.02, *p* < 0.001 in high BA vs. reference low BA) than in boys (OR = 1.36, 95% CI = 0.78–2.39, *p* > 0.05 in moderate BA and OR = 2.15, 95% CI = 1.03–4.47, *p* < 0.05 in high BA vs. reference low BA, respectively).

## 4. Discussion

In the present study, we assessed the relationships between BA and health-related lifestyle in a large sample of adolescent girls and boys. We hypothesised that the health-related lifestyle and well-being of adolescents of both genders would be associated with more positive body image, independent of BMI. The findings of our study confirmed this hypothesis and showed that positive body image (operating as BA) is an important correlate of health-related lifestyle and psychosocial well-being in girls and boys, independent of their body weight. Results of previous studies have suggested that BA is an important indicator of self-esteem, lifestyle-related behaviour and life satisfaction in girls and women ([53]; [54]). However, the findings of our study provide valuable empirical insights, demonstrating that positive body image is also important for the health-related lifestyle and psychosocial well-being of boys.

The analysis of the results for boys showed that the high-BA group reported greater life satisfaction, self-esteem, good/excellent self-rated health, more frequent sports participation, more positive perceived physical fitness, lower disordered eating, lower tobacco smoking and alcohol consumption than boys with lower BA. Some of these findings are consistent with those of previous studies of boys, which indicated that BA is associated with higher self-esteem ([40]; [50]), sports participation and physical activity ([41]; [43]; [69]), lower disordered eating ([9]; [66]) and healthier BMI ([90]).

The present study adds important new empirical data that boys with high BA express more adequate body weight estimation compared to adolescents in lower BA categories. It is critical for adolescents to possess an accurate perception of their body weight, as this facilitates the adoption of beneficial dietary and physical activity behaviours that are integral to the maintenance of a healthy lifestyle ([38]). Many adolescent boys tend to underestimate their weight ([46]), which negatively impacts their weight management strategies, leading to drive for muscularity, disordered eating and unhealthy muscle-gaining behaviours ([27]; [65]; [75]). Therefore, promoting BA may play a key role in supporting boys’ accurate body weight perception and, in turn, protecting them from maladaptive behaviours related to weight and appearance.

Having a positive body image can act as a protective factor, fostering trust, appreciation, respect and care for one’s body. It can also help adolescents to resist appearance-related societal pressures ([7]). According to the sociocultural theory of negative body image ([76]), these pressures contribute to body dissatisfaction and self-objectification ([79]), which can lead to harmful outcomes such as disordered eating. For boys and men, sociocultural messages that promote muscular appearance ideals have been consistently linked to drive for muscularity, lower self-esteem and unhealthy behaviours, including risky activities, disordered eating, and excessive or harmful exercise patterns ([22]; [51]; [74]; [77]). Drive for muscularity was found to be prospectively related to dieting and binge drinking in young men ([24]).

Developing a positive BA may therefore help adolescent boys resist these pressures, cultivate a more functional body image (as opposed to self-objectification), and avoid engaging in unhealthy behaviours such as smoking and alcohol consumption. Our findings suggest that higher BA is associated with lower rates of smoking and alcohol use in boys, indicating that BA may act as a protective factor. However, research on the links between BA, smoking, and alcohol consumption in boys remains limited, and further studies are needed to explore these associations in different samples.

Insights from the Acceptance Model of Intuitive Eating ([7]), which has been studied primarily in girls and women, are also relevant here. This model suggests that when girls and women feel their bodies are unconditionally accepted by important others, they develop a holistic appreciation of their body—valuing its functionality (what it can do) rather than focusing solely on appearance ([1]). Such appreciation fosters gratitude, respect, and care for the body, which may help buffer against health-damaging behaviours like smoking and alcohol use. However, the present study was not fully grounded in the Acceptance Model of Intuitive Eating, and comparable research on boys is still lacking.

For adolescent girls, high BA was related to almost all analysed variables except alcohol consumption. These findings are in line with previous studies reporting that BA is related to greater psychological health including self-esteem and life satisfaction ([54]; [64]; [67]). Our findings in girls also replicated previous studies reporting that BA is related to lower DE ([5]; [57]; [66]), greater self-perceived physical health ([64]), engagement in sports ([41]; [43]; [64]), perceived physical fitness ([40]) and lower tobacco use ([61]). These associations may be partly explained by the Acceptance Model of Intuitive Eating, which highlights the connection between positive body image and a more greater body functionality ([7]). In this framework, BA fosters greater valuing of body functionality—appreciating, respecting, and honouring the body for what it is capable of doing—an orientation that has been linked to more adaptive eating patterns and higher overall well-being ([55]).

Another explanatory mechanism may come from self-objectification theory. Girls who are less likely to prioritize appearance over body functionality, or to monitor their bodies from an outsider’s perspective, may experience lower stress and higher self-esteem. In contrast, greater self-objectification in adolescent girls has been associated with dysfunctional eating patterns and poorer cognitive performance ([20]). Importantly, physical activity may serve as a protective factor, as it can encourage a stronger focus on body functionality and thereby support healthier outcomes ([20]).

Our study adds the important new data that BA is associated with a lower overestimation of body weight which is an important contributor to DE, emotional problems and increased body weight gain over time ([32]; [45]). It seems that girls that have more positive body image has more realistic understanding about their body weight. These findings highlight the importance of fostering BA in preventive interventions, as it may not only reduce the risk of disordered eating and related emotional difficulties but also support healthier weight regulation over time.

In addition, our study provides valuable new insights into the relationship between higher levels of BA in girls and healthier sleep patterns. These associations may reflect better overall mental health and well-being, which have consistently been linked to BA in women and girls ([54]). Girls with greater BA may experience lower stress and higher life satisfaction ([64]), be less prone to monitoring their appearance from an outsider’s perspective, and thus have more cognitive resources for academic activities. They may also derive greater enjoyment from, and show higher participation in, sports—factors that together contribute to improved sleep quality ([48]). This novel finding underscores the potential of BA-focused interventions not only to promote healthier eating and physical activity but also to support sleep regulation.

Finally, we found that in girls, time spent on screen-based activities was negatively associated with BA. Previous research indicates that much of girls’ online time is spent on social media (84), which has been linked to greater body image concerns ([84]). This suggests that girls with higher levels of BA may be less drawn to social media use, or may engage with it in healthier ways, thereby reducing potential risks to their body image.

The findings suggest that while higher BA is beneficial for both genders, the specific pathways and strengths of these associations differ. In girls, BA was more strongly related to healthier sleep, lower screen time, and reduced engagement in disordered or unhealthy eating behaviours, consistent with theories highlighting the protective role of BA against self-objectification and sociocultural appearance pressures. In boys, BA was uniquely associated with abstaining from alcohol, suggesting that positive body image may buffer against risk behaviours more strongly in this group. The interaction effects further indicate that BA exerts a stronger influence on self-esteem, BMI, and disordered eating behaviours in girls than in boys. This may reflect the heightened sociocultural pressures on girls’ appearance and/or the greater vulnerability of girls to body-related concerns, meaning that BA plays a particularly crucial role in promoting their psychological well-being and resilience.

### 4.1. Strengths and Limitations

A key strength of this study is the large sample of boys and girls, which allowed for a comprehensive assessment of the relationship between BA and lifestyle-related factors while controlling for BMI. The findings carry important implications for policy and practice, suggesting that health education and promotion programs for adolescents may be more effective when they explicitly incorporate a focus on positive body image. While encouraging adolescents to adopt healthy behaviours is essential, it is equally important to educate them about BA—fostering comfort and acceptance of their bodies regardless of weight. Such education should emphasize self-respect, attentiveness to bodily needs, acceptance of imperfections, and resilience against sociocultural pressures. Overall, the results of this study support a theoretical framework that prioritizes inclusive, health-focused education and environments designed to promote well-being ([11]; [59]; [62]; [80]).

### 4.2. Practical Implications

These findings highlight the importance of promoting both BA and healthy lifestyle behaviours in adolescents. They underscore the need for future research to examine whether fostering BA can encourage healthier habits and, conversely, whether promoting healthy lifestyle behaviours can strengthen more positive body image. While body weight management remains an important focus in health education, it should be approached in ways that avoid body weight stigma and support positive body image. Interventions should be tailored to gender-specific needs, emphasizing prevention of dysfunctional eating for girls, and reducing substance use for boys. Incorporating body-positive education into schools and health programs can help create supportive environments that enhance both psychological well-being and physical health.

## 5. Conclusions

The findings of the present study expand previous results assessing the role of positive body image for health-related lifestyles in adolescent girls and provide new empirical data for adolescent boys. The results of this study showed that higher BA was significantly associated with greater life satisfaction, higher self-esteem, better perceived physical fitness and healthier eating behaviours, independent of BMI, in boys and girls. Adolescents with high BA had significantly higher odds of participating in sport, having a healthy BMI, accurate body weight estimation, good self-rated health and non-smoking behaviours compared to those with low BA. BA was also associated with healthier sleeping hours, lower screen time and lower unhealthy and DE behaviour in girls. Boys with high BA were more likely to abstain from alcohol. The interaction effects indicated that the effect of BA on self-esteem, BMI and DE behaviours was stronger in girls than in boys.

## Figures and Tables

**Table 1 behavsci-15-01400-t001:** Sample characteristics.

Study Measures		
Body appreciation, n (%)	low (<−1 SD)	269 (19.1)
	moderate (mean ± 1 SD)	843 (59.7)
	high (>1 SD)	300 (21.2)
Body appreciation, m (SD)		3.31 (1.14)
Self-esteem, m (SD)		28.55 (6.16)
Life satisfaction, m (SD)		7.19 (2.26)
Body mass index, n (%)	underweight	167 (11.8)
	healthy	1065 (75.4)
	overweight/obese	180 (12.8)
Body mass index kg/m^2^, m (SD)		21.37 (3.07)
Body weight estimation	underestimation	180 (12.7)
	adequate	929 (65.8)
	overestimation	303 (21.5)
Self-rated health, n (%)	poor	65 (4.6)
	fair	388 (27.5)
	good	664 (47.0)
	excellent	295 (20.9)
Participation in sports	no	346 (24.5)
	yes	1066 (75.5)
LTEQ score, m (SD)		67.78 (44.49)
Perceived physical fitness, m (SD)		2.10 (1.01)
Sleep duration, hours/day, m (SD)		7.22 (1.33)
Time playing computer games, hours/day, m (SD)		1.02 (1.66)
Time browsing internet, hours/day, m (SD)		4.10 (2.31)
Unhealthy eating habits, m (SD)		1.76 (0.63)
EDE-Q 6 total score, m (SD)		1.38 (1.24)
Smoking, n (%)	never	972 (68.8)
	less than once/week	137 (9.7)
	once/week	111 (7.9)
	everyday	192 (13.6)
Alcohol intake until feeling dizzy n, (%)	never	511 (36.2)
	once	191 (13.5)
	2–3 times	266 (18.8)
	4–10 times	201 (14.2)
	>10 times	243 (17.2)

Note. n—number of study participants, %—percentage, SD—standard deviation, LTEQ—Leisure-Time Exercise Questionnaire, EDE-Q 6—Eating Disorder Examination Questionnaire 6.

**Table 2 behavsci-15-01400-t002:** Comparison of the continuous study measures (m ± SE) across low-, moderate- and high-body appreciation groups in boys with the body mass index as a covariate.

Study Measures	Low BA, n = 100	Moderate BA, n = 343	High BA, n = 127	Partial ŋ^2^	*p*
Body mass index, kg/m^2^	22.52 ± 0.30	21.96 ± 0.16	21.65 ± 0.26	—	0.092
Life satisfaction	6.22 ± 0.23	7.50 ± 0.12 ^a^	8.59 ± 0.20 ^ab^	0.10	<0.001
Self-esteem	25.35 ± 0.56	28.53 ± 0.30 ^a^	32.17 ± 0.50 ^ab^	0.13	<0.001
Physical activity score	78.42 ± 4.57	75.19 ± 2.46	82.30 ± 4.05	—	0.314
Perceived physical fitness	2.35 ± 0.10	2.16 ± 0.05	2.73 ± 0.09 ^ab^	0.05	<0.001
Sleep, hours/day	7.35 ± 0.14	7.43 ± 0.08	7.50 ± 0.12	—	0.730
Internet browsing, hours/day	3.89 ± 0.23	3.83 ± 0.13	3.63 ± 0.21	—	0.646
Computer games, hours/day	2.04 ± 0.20	1.90 ± 0.11	1.78 ± 0.17	—	0.617
Unhealthy nutrition score	1.76 ± 0.07	1.68 ± 0.04	1.65 ± 0.06	—	0.413
Disordered eating	1.04 ± 0.08	0.93 ± 0.04	0.60 ± 0.07 ^ab^	0.04	<0.001

Note. m—mean, SE—standard error, ŋ^2^—eta-squared, BA—body appreciation; ^a^ *p* < 0.05 compared to low BA, ^b^ to moderate BA.

**Table 3 behavsci-15-01400-t003:** Comparison of the continuous study measures (m ± SE) across low-, moderate- and high-body appreciation groups in girls with the body mass index as a covariate.

Study Measures	Low BA, n = 169	Moderate BA, n = 500	High BA, n = 173	Partial ŋ^2^	*p*
Body mass index, kg/m^2^	22.24 ± 0.23	20.90 ± 0.13 ^a^	19.88 ± 0.23 ^ab^	0.06	<0.001
Life satisfaction	5.17 ± 0.15	7.09 ± 0.09 ^a^	8.33 ± 0.15 ^ab^	0.21	<0.001
Self-esteem	23.36 ± 0.42	28.21 ± 0.24 ^a^	33.80 ± 0.41 ^ab^	0.27	<0.001
Physical activity score	58.51 ± 3.29	60.42 ± 1.88	66.60 ± 3.24	—	0.171
Perceived physical fitness	1.69 ± 0.08	1.95 ± 0.04 ^a^	2.21 ± 0.07 ^ab^	0.03	<0.001
Sleep, hours/day	6.74 ± 0.10	7.12 ± 0.06 ^a^	7.27 ± 0.10 ^a^	0.02	<0.001
Internet browsing, hours/day	4.65 ± 0.18	4.30 ± 0.10	3.98 ± 0.18 ^a^	0.01	0.032
Computer games, hours/day	0.69 ± 0.08	0.40 ± 0.05 ^a^	0.28 ± 0.08 ^a^	0.02	0.001
Unhealthy nutrition score	1.85 ± 0.05	1.85 ± 0.03	1.70 ± 0.05 ^b^	0.01	0.018
Disordered eating	2.59 ± 0.09	1.74 ± 0.05 ^a^	0.82 ± 0.09 ^ab^	0.20	<0.001

Note. m—mean, SE—standard error, ŋ^2^—eta-squared, BA—body appreciation; ^a^ *p* < 0.05 compared to low BA, ^b^ to moderate BA.

**Table 4 behavsci-15-01400-t004:** Odds ratios of healthy lifestyle factors according to body appreciation in boys and girls with body mass index as a covariate.

Study Measures	Sex	Low BA	Moderate BA		High BA	
		Ref.	OR	95% CI	OR	95% CI
Healthy body mass index (1) vs. overweight/obesity (0)	boys	1.00	1.36	0.78–2.39	2.15 *	1.03–4.47
	girls	1.00	2.40 ***	1.50–3.87	8.78 ***	3.35–23.02
Adequate body weight estimation (1) vs. overestimation (0) ^a^	boys	1.00	1.48	0.67–3.26	7.54 *	1.59–35.64
	girls	1.00	2.45 ***	1.67–3.60	7.22 ***	4.20–12.39
Good/excellent self-rated health (1) vs. average/poor health (0)	boys	1.00	1.36	0.84–2.21	4.10 ***	2.01–8.36
	girls	1.00	2.17 ***	1.52–3.12	5.93 ***	3.58–9.80
Participation in sports (1) vs. non-participation (0)	boys	1.00	1.44	0.83–2.48	2.53 *	1.21–5.33
	girls	1.00	1.55 *	1.06–2.25	1.76 *	1.10–2.83
Non-smoking (1) vs. smoking (0)	boys	1.00	1.98 **	1.23–3.20	1.89 *	1.06–3.36
	girls	1.00	1.41	0.92–2.16	1.87 *	1.05–3.31
No alcohol consumption (1) vs. alcohol consumption (0)	boys	1.00	1.81 *	1.14–2.89	1.99 *	1.16–3.43
	girls	1.00	1.04	0.73–1.48	1.51	0.97–2.35

Note. Ref.—reference group, OR—odds ratio, CI—confidence interval, BA—body appreciation, * *p* < 0.05, ** *p* < 0.01, *** *p* < 0.001, ^a^ adolescents in the underestimation category for this analysis were set as missing values and excluded.

## Data Availability

The original contributions presented in this study are included in the article. Further inquiries can be directed to the corresponding author.

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
