# Peer review of "Exploring the Link Between Body Appreciation and Health-Related Lifestyle in Adolescents: A Cross-Sectional Study"

_behavsci, 2025, doi:10.3390/bs15101400_

Round 1
Reviewer 1 Report
Comments and Suggestions for Authors
Dear Authors,
Thank you for providing the opportunity to read this interesting paper. Below, I have listed my comments:
1) In the introduction, it is stated that there is a research gap in adolescent boys, but the discussion is still heavily weighted toward girls. It would have been nice to specify if any of the studies reviewed have looked into boys so that you can ensure equal conceptual treatment across genders
2)in the results, the interaction effects (sex × BA) are mentioned briefly without detail or accompanying figures/tables. Consider including a table or figure to illustrate the interaction effects, especially for variables with strong sex differences (e.g., self-esteem, BMI). If omitted for brevity, clearly explain the direction of interaction.
3) In the discussion, while many associations are mentioned, the text lacks theoretical integration. There’s little engagement with why BA is protective and no reference to psychological mechanisms (e.g., self-objectification theory, body functionality appreciation etc).
4) It would also be good to emphasize unique insights. New contributions (e.g., BA’s link to accurate weight perception, sleep in girls) are mentioned but not deeply explored. Explain why they matter: e.g., This finding is novel and underscores the potential of BA interventions to target not only physical activity or eating but also sleep regulation and media use.
5) Lastly, while differences between boys and girls are noted, the interaction effects are not sufficiently interpreted. how and why BA impacts girls more strongly in certain domains (e.g., internalization of appearance norms, higher prevalence of DE)?
I hope this feedback is helpful.
Author Response
Dear Authors,
Thank you for providing the opportunity to read this interesting paper.
Thank you for reviewing our article and for your positive feedback. All changes made in the text are in a blue font.
Below, I have listed my comments:
1) In the introduction, it is stated that there is a research gap in adolescent boys, but the discussion is still heavily weighted toward girls. It would have been nice to specify if any of the studies reviewed have looked into boys so that you can ensure equal conceptual treatment across genders
Thank you for this comment. We have improved the introduction to ensure equal representation of both genders.
2)in the results, the interaction effects (sex × BA) are mentioned briefly without detail or accompanying figures/tables. Consider including a table or figure to illustrate the interaction effects, especially for variables with strong sex differences (e.g., self-esteem, BMI). If omitted for brevity, clearly explain the direction of interaction.
Thank you for this important comment. We highlighted in the results section these interaction effects.
3) In the discussion, while many associations are mentioned, the text lacks theoretical integration. There’s little engagement with why BA is protective and no reference to psychological mechanisms (e.g., self-objectification theory, body functionality appreciation etc).
Thank you for this comment. We have rewritten the discussion to include paragraphs that explain the possible mechanisms behind the associations.
4) It would also be good to emphasize unique insights. New contributions (e.g., BA’s link to accurate weight perception, sleep in girls) are mentioned but not deeply explored. Explain why they matter: e.g., This finding is novel and underscores the potential of BA interventions to target not only physical activity or eating but also sleep regulation and media use.
We have expanded the discussion on novel findings by rewriting the discussion and extending the paragraphs.
5) Lastly, while differences between boys and girls are noted, the interaction effects are not sufficiently interpreted. how and why BA impacts girls more strongly in certain domains (e.g., internalization of appearance norms, higher prevalence of DE)?
Thank you for this comment. We included possible explanation based on the theory.
I hope this feedback is helpful.
Thank you.
Reviewer 2 Report
Comments and Suggestions for Authors
Abstract –
It would useful (in a few additional words) within the opening sentence to contextualise the rationale the project alongside the aims. The latter is clear but the “why” behind the project would help highlight the importance of the project more. Otherwise the section is clear and provides a general, good coverage. If it is possible to summarise the key/salient findings in a more concise fashion, this would be advantageous, but this shouldn’t be a “mandatory” change and more a suggestion. As the authors have noted the full analysis plan here, it does make sense to details all of the results.
Introduction –
Generally, this is well written (a few structural issues highlighted below) and does stress the importance of the topic generally. There is a good literature review present. The main focus for improvement is why this cross sectional study in BA is needed specifically and why the focus on sex/gender. This is hinted at in L85-87, but this does need developed a little more for the readers. This will also strength the manuscript in other areas too.
There is certainly some interesting scope to develop further in terms of L65-70. Body image concerns are associated with healthier habits/behaviours, but there are downsides to this too (and it doesn’t per say mean you will not engage in risky health behaviours as a result). BA, however, is offering a buffer for negative health behaviours? Cognitive dissonance might be a good route here?
L 33-38: do these sentences flow on naturally? Perhaps “growth” could be substituted for “change” as it’s the change that might contextualise why body positivity would naturally decline. Perhaps line 34-35 would be better placed underneath / at the end of this. I.e., teenagers experience changes and unhealthy health behaviours which in turn challenges to positive body image?
L51: “screen–based”
I like the use of the sex/gender difference to identify why BA is worth exploring further, however a bit more coverage of “why” this might be the case might make the discussion/results easier to interpret later. This seems a rather important aspect to the study given the results/discussion and therefore should be given the appropriate context.
Method –
Inclusion criteria seem a little small for a study of this nature. Typically, it would have been advantageous to see some additional exclusion (for participant protection and data integrity) for eating disorder diagnoses/symptomology, given the nature of the scales being used. Could the authors potentially add some further ethical information here? For example, were participants signed posted to supporting resources? What was the procedure if a participant showed some troubling high scores on one or more of these scales?
Could the materials be better subtitled to make it easier for the reader to follow, please?
Analysis plan is clear. However, is it worth number hypotheses in the introduction and then making it expliticity clear in the method how each analysis addresses which hypothesis? There is a lot of analysis here (which is fine, if there is a clear hypothesis for each), but some considerations of Type I errors would be worthwhile here.
Results –
Is there a rationale for using ± 1 SD for the groupings of BA? This should be made clear.
Perhaps some of these tables could be supplementary materials, quite hard to follow the absolute “salient” information at the moment.
Again, with sheer number of separate analyses conducted – could there not be some additional correction applied?
Discussion –
This section perhaps needs the largest amount of work. The authors have done well to explain how the results of this study sit with previous literature, but there is a lack of organisation of what the key findings are. This is a large study with a lot of data and analysis, any findings reported in the results should be explored in full here. Perhaps going back to the numbered hypotheses idea, could this help plan and structure this section better?
The conclusions seem a little off given what the focus of the analysis is. These could be far more specific. There has been a large focus on sex/gender as a variable and this should inform the conclusions and the implications when suggesting that this could support education and policy.
There is perhaps some contradictory messaging in the discussion of results and limitations aspect of this section. A couple of examples would be: in an earlier sentence, there is claim that the study is supporting the education of BA in schools in X, Y and Z manner but then later states that the results are limited? Moreover, the results apparently support the use of BA to enhance wellbeing, but then mental health wasn’t measured making this a redundant comment. Perhaps note strengths and limitations and then weight them up for concluding remarks and implications, as both impact?
The limitations are glossed over when these should be explained in full. Why should this be crossed culturally confirmed (any literature to support this)? Cross-section design seemed intentional, this could be explored further (time/dates/seasons that the data collection took place might be noteworthy for example).
Additional comments:
For the Informed Consent Statement at the end of the manuscript, this should be updated to reflect that both the participant and their parent provided consent?
Author Response
Review 2
Thank you for reviewing our manuscript and for your valuable comments. All changes have been marked in blue in the revised manuscript.
Abstract –
It would useful (in a few additional words) within the opening sentence to contextualise the rationale the project alongside the aims. The latter is clear but the “why” behind the project would help highlight the importance of the project more. Otherwise the section is clear and provides a general, good coverage. If it is possible to summarise the key/salient findings in a more concise fashion, this would be advantageous, but this shouldn’t be a “mandatory” change and more a suggestion. As the authors have noted the full analysis plan here, it does make sense to details all of the results.
Thank you, we included opening sentence as suggested.
Introduction –
Generally, this is well written (a few structural issues highlighted below) and does stress the importance of the topic generally. There is a good literature review present. The main focus for improvement is why this cross sectional study in BA is needed specifically and why the focus on sex/gender. This is hinted at in L85-87, but this does need developed a little more for the readers. This will also strength the manuscript in other areas too.
Thank you. The rationale for the study has been expanded to include a new paragraph.
There is certainly some interesting scope to develop further in terms of L65-70. Body image concerns are associated with healthier habits/behaviours, but there are downsides to this too (and it doesn’t per say mean you will not engage in risky health behaviours as a result). BA, however, is offering a buffer for negative health behaviours? Cognitive dissonance might be a good route here?
Thank you for this comment. We found a mistake in the text: 'body image concerns are associated with' should be changed to 'positive body image is associated with'. It's possible that this happened during the editing process. We have revised this paragraph.
L 33-38: do these sentences flow on naturally? Perhaps “growth” could be substituted for “change” as it’s the change that might contextualise why body positivity would naturally decline. Perhaps line 34-35 would be better placed underneath / at the end of this. I.e., teenagers experience changes and unhealthy health behaviours which in turn challenges to positive body image?
Thank you for the comment, but we think that the text ok. “Adolescence is a period characterised by significant physical, psychological and social changes that affect the adolescent's growth [1]. These changes can pose challenges for developing and/or maintaining positive body image [2].
L51: “screen–based”
Thank you, corrected.
I like the use of the sex/gender difference to identify why BA is worth exploring further, however a bit more coverage of “why” this might be the case might make the discussion/results easier to interpret later. This seems a rather important aspect to the study given the results/discussion and therefore should be given the appropriate context.
Thank you. We have revised the introduction and provided a clearer justification for the study.
Method –
Inclusion criteria seem a little small for a study of this nature. Typically, it would have been advantageous to see some additional exclusion (for participant protection and data integrity) for eating disorder diagnoses/symptomology, given the nature of the scales being used. Could the authors potentially add some further ethical information here? For example, were participants signed posted to supporting resources? What was the procedure if a participant showed some troubling high scores on one or more of these scales?
Could the materials be better subtitled to make it easier for the reader to follow, please?
We thank the reviewer for highlighting the importance of participant protection and ethical safeguards. Our study was conducted with a non-clinical adolescent sample recruited from the general population. We did not apply exclusion criteria related to eating disorder diagnoses or symptomatology, as the study aimed to capture a broad and ecologically valid picture of adolescents’ positive body image and lifestyle factors.
The participants were introduced to the survey content prior to it and could express their willingness to participate or decline it. Also, participants could discontinue the survey at any time by closing the browser, in which case their responses were not recorded. No clinical follow-up or signposting to external support services was provided since the study had no aims to do that.
We added this information to the methods section (2.1 Study design and procedure).
Also, we subtitled materials.
Analysis plan is clear. However, is it worth number hypotheses in the introduction and then making it expliticity clear in the method how each analysis addresses which hypothesis? There is a lot of analysis here (which is fine, if there is a clear hypothesis for each), but some considerations of Type I errors would be worthwhile here.
Thank you for this comment. In the current manuscript, we applied post hoc analyses and controlled for multiple comparisons to reduce the risk of Type I error. We have also expanded the Statistical Analysis section to further explain how Type I error was controlled.
On the other hand, we avoided excessive simplification by not expanding on the hypotheses, but instead discussed the results in more detail in the Discussion section, as recommended.
Results –
Is there a rationale for using ± 1 SD for the groupings of BA? This should be made clear.
There are some previous studies made on ± 1SD grouping in case grouping criteria or population norms are missing. These authors, including our previous studies, used this grouping. We included this explanation to the Methods section.
Homan, K. J., & Tylka, T. L. (2015). Self-compassion moderates body comparison and appearance contingent self-worth’s inverse relationships with body appreciation. Body Image, 15, 1-7. https://doi.org/10.1016/j.bodyim.2015.04.007
Wodarz, R.; Rogowska, A.M. The Moderating Effect of Body Appreciation on the Relationship between Self-Esteem and Life Satisfaction. Eur. J. Investig. Health Psychol. Educ. 2024, 14, 870-887. https://doi.org/10.3390/ejihpe14040056
Jankauskiene, R., & Baceviciene, M. (2022). Media Pressures, Internalization of Appearance Ideals and Disordered Eating among Adolescent Girls and Boys: Testing the Moderating Role of Body Appreciation. Nutrients, 14(11), 2227. https://doi.org/10.3390/nu14112227
Perhaps some of these tables could be supplementary materials, quite hard to follow the absolute “salient” information at the moment.
We appreciate the reviewer’s suggestion regarding the presentation of tables. We agree that streamlining the results would help readers to focus on the most salient findings. However, in the revised manuscript, we decided to retain these tables as the main results because they directly illustrate our main findings.
Again, with sheer number of separate analyses conducted – could there not be some additional correction applied?
Thank you for this important comment. We applied post hoc analyses and controlled for multiple comparisons to reduce the risk of Type I error. We added this information to the statistical analysis section.
Discussion –
This section perhaps needs the largest amount of work. The authors have done well to explain how the results of this study sit with previous literature, but there is a lack of organisation of what the key findings are. This is a large study with a lot of data and analysis, any findings reported in the results should be explored in full here. Perhaps going back to the numbered hypotheses idea, could this help plan and structure this section better?
Thank you for this comment. We decided not to develop hypotheses as there would be more than five of them. Instead, based on the comments of the first reviewer, we provided a theoretically grounded explanation of the results and expanded discussion on novel findings.
The conclusions seem a little off given what the focus of the analysis is. These could be far more specific. There has been a large focus on sex/gender as a variable and this should inform the conclusions and the implications when suggesting that this could support education and policy.
Thank you. We have revised the conclusions to provide more specific practical implications.
There is perhaps some contradictory messaging in the discussion of results and limitations aspect of this section. A couple of examples would be: in an earlier sentence, there is claim that the study is supporting the education of BA in schools in X, Y and Z manner but then later states that the results are limited? Moreover, the results apparently support the use of BA to enhance wellbeing, but then mental health wasn’t measured making this a redundant comment. Perhaps note strengths and limitations and then weight them up for concluding remarks and implications, as both impact?
The limitations are glossed over when these should be explained in full. Why should this be crossed culturally confirmed (any literature to support this)? Cross-section design seemed intentional, this could be explored further (time/dates/seasons that the data collection took place might be noteworthy for example).
Thank you for these comments. We have revised the limitations accordingly.
Additional comments:
For the Informed Consent Statement at the end of the manuscript, this should be updated to reflect that both the participant and their parent provided consent?
Thank you, this information is now added.
Round 2
Reviewer 1 Report
Comments and Suggestions for Authors
Thank you Authors. Good luck with the rest of the process.
Author Response
We thank the reviewer.